# Emotional Response in the Elderly to Sports Materials

**DOI:** 10.3390/ijerph191811775

**Published:** 2022-09-18

**Authors:** Juan Carlos Fernández-Truan, Cristina María Machado Arenós

**Affiliations:** 1Departamento del Deporte e Informática, Universidad Pablo de Olavide de Sevilla, 41013 Sevilla, Spain; 2Consejería de Educación Junta de Andalucía, CEIP, Júpiter, 41300 San José de la Rinconada, Spain

**Keywords:** education, health, emotions, materials, physical education, emotional intelligence, emotional education, elderly

## Abstract

This research aims to understand the effectiveness of the material resources used by 365 elderly people who perform physical activities in Seville (Spain), based on the sensations that the use of these objects gave them. A descriptive design was used through a questionnaire validated by the Delphi method with 10 pre-selected experts. The results showed the existence of significant relationships between the emotions generated in the elderly and the handling of different materials. This is essential for the planning of physical activities for this type of user in order to achieve greater motivation and continuity.

## 1. Introduction

Since the mid-twentieth century, there has been a marked increase in the number of people over the age of 65 in developed countries and their numbers are expected to continue to rise. This is leading to the progressive ageing of the population at the European level. According to the National Institute of Statistics (SIN), in Spain, in the early twentieth century, the population above 65 years old was 5.3% of the total population; in the 1950s, it was 9–10%, and currently, it is about 15.8% of the Spanish population, and it is expected that in the near future, the total will be 25%. The SIN states that Spain’s population is expected to grow progressively over the next 40 years, but this will be accompanied by an ageing population.

Because of this, since the late twentieth century, numerous research studies have been performed on elderly people and their quality of life, making it a top topic in both conferences and published articles, as well as an area of training for many professionals from different scientific disciplines. Research of more relevant ones that help to identify concerns and unravel risk factors for health, together with life quality improvement for the elderly, definitely leads to a real challenge in our current society. However, research needs to explain how other factors affect the elderly’s everyday actions. Hence, there is a big effort to identify the habits, behaviours, attitudes, and interests of the elderly.

Our research represents another step towards that study line, focused on the efficiency of the physical activities performed by the elderly community in sports centres seconded to Seville SMIs. The main input in our research is the materials used looking for those that are found to be more interesting for the elderly. The aim is to discover which emotions are awakened by these materials and resources based on the differences between weight, size, type, colour, texture, etc. Only one characteristic will remain the same: the shape. This will always be spherical. For this reason, the novelty in our research is to link the three elements we consider relevant for the development of this type of physical activity: the users (the elderly), the medium (specific resources), and the responses (emotions aroused by the resources). The results obtained may give relevant information for the selection of the resources used in the sports sessions based on each context; additionally, identifying which resource draws more attention from the users. A feeling of security and motivation in the users will help them be consistent with physical activity and reach a greater quality of life.

In that aspect, all elements involved in the application of physical-sportive activities with the elderly are not only looking into making sports more attractive but also safer as a primary factor. The use of mobile objects can represent a considerable risk that must be analysed very carefully in order to avoid any inadequate use.

Research on the use of sportive objects for the elderly has been almost non-existent so far.

Throughout history, many philosophers and thinkers, such as Plato, Aristotle, empiricists, agnostics, Christian philosophers, etc. have given great importance to the knowledge of the senses. However, some researchers, such as the modern rationalists (Descartes, Spinoza, Leibniz), declare that since the beginning of time, the senses could be an element of reality deceit. This varies depending on the time and era in which these philosophers lived.

Nowadays, it is known that through the senses, human beings have the capacity to send data to the brain, where the information is displayed as emotion. When this information travels to the brain, it reaches the internal sensitive receivers, which are: the mechanoreceptors, the thermoreceptors, the nanoreceptors, the electromagnetic receptors, and the chemoreceptors. The most relevant external receptors for this research are sight, hearing, and touch. They are in charge of sending the information coming from the exterior through stimulation: light, sound, and pressure [1].

Through the internal and external sensitive receptors, we receive sensations, known for their conscious process. Sensations are the immediate result of the excitation of the organs through which the human being knows of the existence and properties of specific objects [2]. They are also considered an immediate response by the sensory organs due to a stimulus [3]. Through sensations, perceptions are generated, defined as ‘the inner feeling that comes from a material stimulus through sense organs’ [4]. At the same time, perception is linked to emotions, defined as the condition of intense affection, normally originated by a situation, thought, or a pleasant or unpleasant image, which alerts and stirs up the subject. Therefore, an emotion is: ‘the state of consciousness, however meaning pleasant or painful, originated with endogenous or external stimuli, and often accompanied by precise and objective physical changes’ [5].

Emotions have been researched by Paul Ekman [6] and other researchers over the years. However, for our study, we have focused on the emotions mentioned by Ekman [7], as he is a pioneer psychologist in the study of feelings, emotions, and their relationship with facial expressions. The analysed emotions were:-Disgust: From a sociological and cultural and historical perspective, disgust is defined as ‘a social mechanism conditioned by culture and transmitted by pedagogy, which uses facial reflexes and nausea to preserve basic social identity acquired in a prerational way’ [6].-Happiness: It is ‘fulfilling our expectations, desires and projects, which causes a positive feeling, accompanied by the impression of lightness, and widening mood’ [7].-Anger. It is an emotion that arises when the issues we consider important in our life have been violated by someone, including by the person themselves. It was described by St. Tomás de Aquino as an ‘appetite for revenge with incandescence of the body’ [8].-Fear: It is ‘the mood disturbance because of a damage that really threatens or is created by imagination’ [8].-Surprise: It is considered ‘the perception of something new, strange, or something that appears suddenly, causing a brief feeling, which can be negative or positive, which focuses attention on the perceived’ [7].-Sadness: Considered as ‘a loss, a misfortune, a setback that makes it impossible for the realization of our desires or projects and causes a negative feeling, accompanied by a desire to move away from isolation and passivity’ [7].

## 2. Intelligence and Emotional Intelligence

For an adequate identification of these emotions, it is necessary to start from the concept of Emotional Intelligence, but this cannot be understood without explaining first the concept of intelligence itself. The term Intelligence has had many connotations, evolving its definition through the years by various authors. It comes from the Latin ‘*intellegere*’, a composition of two words: ‘*inter*’ (between), and ‘*legere*’ (read, choose). Etymologically speaking, intelligent means ‘he who knows how to choose’, which means that intelligence is the thing that allows people to choose the best option to resolve an issue. Nevertheless, defining intelligence has always been a subject of controversy due to the multitude of opinions given by the authors who analysed it, therefore showing the complexity of the said term. Following Vernon (1960) [9], it was suggested a classification based on three definition groups: psychological, biological, and operatives.

The first one shows intelligence as a cognitive capacity, of learning, and interrelationship; the second considers it as a capacity of adjusting to new situations; and the third gives a circular definition, declaring intelligence as ‘that which the tests measure the intelligence’ [9]. Other definitions are collected by Castejón [10] as: ‘Intelligence is the capacity to acquire aptitude’. According to Woodrow; ‘intelligence is what is measured by the intelligence tests’.

In general terms, the ‘Real Academia Española’ (Royal Spanish Academy) currently defines intelligence as the capacity to understand, comprehend, and invent. Likewise, the concept of intelligence contains different characteristics: the ability to reason, plan, think abstractly, solve problems, understand ideas and languages, and learn. At the same time, the Mainstream Science on Intelligence established a definition of intelligence signed in 1994 by fifty-two researchers all around the world which conceptualized it as: ‘A very general mental capacity, which, among other things, involves the ability to reason, plan, solve issues, think abstractly, understand complex ideas, learn fast and learn from experiences. It is not just a simple book learning, nor a strictly academical ability, or a talent to overcome a test. Rather, the concept refers to the capacity to comprehend the environment itself’ [11].

During the late twentieth century, various psychological theories gained great relevance, which are considered still valid nowadays:

(a) The Intelligence Triarchic Theory, by Robert J. Sternberg [12]; establishes three types: analytical-componential, experiential-creative, and contextual-practice.

(b) Multiples Intelligences Theory, by Howard Gardner; a north American psychologist from Harvard University, who wrote in 1983 ‘The Structure of the Mind’, a work which considered the concept of intelligence as a potential that every human being has, to a greater or lesser degree, proposing that it could not be measured by standardized instruments in an IQ test and offered criteria not to measure but to observe and develop it. He defined intelligence as ‘The capacity to solve problems, or create products valued in one or more cultural contexts’ [13]. Therefore, intelligence must be considered as a biological and psychological potential that allows us to process information that can be activated in order to solve problems or create products with a value for a specific culture. Various researchers have shown that intelligence is not unique, but multiple [13] and can be improved upon. Thus, the latest researchers are not as focused on the intellectual potential of a subject, as they are on the way to use and apply said intelligence. Terms like aptitude, capacity, skills, abilities, proficiency, gifts, talents, or competences are, to a large degree, types of intelligence.

If the brain is composed of many aptitudes, it is because multiple intelligences exist and not only one, as Piaget and many of his followers believed. Human development is composed of a variety of domains that are not only formed by logical-mathematics but also contain intelligences like linguistic, visuospatial, bodily-kinaesthetic, musical, logical-mathematical, intrapersonal, interpersonal, and natural.

(c) Emotional Intelligence Theory. To be able to identify emotions and control them, it is necessary to understand Emotional Intelligence, known by Gardner [14] as Intrapersonal and Interpersonal Intelligence.

The term Emotional Intelligence was defined and used for the first time with the name Social Intelligence in 1920 by the psychologist Edward Thorndike, but it would not be until 1995 when the American psychologist Daniel Goleman published his book ‘Emotional Intelligence’ that it acquired world fame, declaring that ‘Emotional Intelligence includes self-control, enthusiasm, persistence and the ability to motivate oneself… There is an old-fashioned word that encapsulates all ranges of skills that makes up the Emotional Intelligence: character’ [15]. He defined it as a way to interact with the world, which has to take into account various factors such as feelings, impulses, self-consciousness, motivation, enthusiasm, and specific social and communication skills; all of them are influenced by the understanding and the expression of the emotions. For Goleman, emotional intelligence was the capacity to recognize the emotions of both ourselves and others and the ability to handle them. This author considered that emotional intelligence can be organised into five capacities: knowing our own feelings and emotions, handling them, recognizing them, creating our own motivation, and handling their relationship. Although, by 1990, Peter Salovey and John Mayer were the ones to coin the Emotional Intelligence term quoted, understanding it as the ability to handle feelings and emotions, discriminate between them, and use this knowledge to direct our own thoughts and actions, defining it as ‘a part of the social intelligence that includes the ability to control our emotions and the rest, discriminate between them and use said information to guide our thoughts and our behaviour’ [16].

Just like ‘the skillset that explains the individual differences in the way to perceive and understand our emotions. In a more formal setting, it is the ability to perceive, value and express emotions with precision, the ability to access and/or generate feelings to ease thinking, to understand emotions and emotional reasons, and finally, the ability to handle emotions both our own and other’s [16].

Attached to Emotional Intelligence, a term that can be usually found is Emotional Education, an essential element for the development of Emotional Intelligence. Without adequate Emotional Education, we can hardly reach correct Emotional Intelligence, thus describing why Emotional Education is understood as an educative ongoing and permanent process. It tries to maximize the development of the emotional competencies of people as an essential element of the development of human beings, with the objective of training for their life and with the finality of increasing their social and personal wellness [17]. Body Intelligence, defined by Piaget (1956) [18], with the name Motor Intelligence, has also been a determinant for this research. It is an intelligence that does not use words, thoughts, or formal logic, but sensory and motor. According to it, sensory is everything that relates to our senses, whereas motor is everything associated with our locomotor system, from our first days of life until the last, showing that no action is completely innocuous with respect to the development of the awareness [18].

## 3. Physical Activities and Ageing

As this research focuses on the use of material resources in elderly people, it is essential to point out the resources that have been selected for this study. It was decided to choose 20 homologated materials and alternatives specific to Physical and Sportive education, where there were different textures, colours, sizes, and weights, only allowing the shape of the element to remain constant. The homologated materials were a table tennis ball, tennis ball, base ball, golf ball, boules’ skittles, billiards ball, rhythmic gymnastics ball, volleyball, and basketball. The non-homologated or alternative materials were an American cork ball, hackey sack ball, cork ball, reflex ball, floorball ball, juggling ball, little tennis ball, little basketball, rubber ball, and fit ball.

When focusing on elderly people, it is necessary to take into account that the education of these people must be very particular and individualized, always keeping in mind the personal and individual characteristics of each group member where the work is being performed. Therefore, it is necessary to know the conceptual differences between the type of activity to be performed: physical activity, physical exercise, or sport [19]. Their different characteristics can be defined as ‘Physical Activity: intentional and voluntary body movements that consume energy and are part of the daily habit of a person’ [19]; or ‘the body movements produced by the skeletal muscles that consume energy’ [20]. Physical Exercise is understood as the ‘type of physical activity where repetitive movement is done, planned and structured, done to improve or keep the physical aptitude of an individual. This includes activities like walking quickly, cycling, aerobics, gymnastics, swimming, etc’ [20]. By sport, it is meant as ‘the physical exercise governed by rules or regulations, and it is done with a competitive finality toward oneself, others or the environment’ [20].

On the other hand, what we understand for ageing is the natural process that naturally occurs throughout life, accompanied by biopsychosocial alterations [21]. However, being an old person is not a synonym for a physically or pathologically ill person, although the risk of having an illness due to the degeneration of the physical, sensory, or cognitive systems increases with age. None of these degradations are produced equally among the elderly. As such, we can find a great diversity and range of degeneration in their performance, making the degree of autonomy for the old people vastly different, which forces us to adjust the development of the physical activities in three sizeable groups:-Independent elders: Those that enjoy a high degree of autonomy and wellness, without any serious pathology that limits their independence.-Moderately dependent elders: Those who suffer from some type of pathology that reduces their mobility and autonomy.-Especially frail elders: Those elders in a situation of dependency suffering from degenerative pathologies that limit their mobility and autonomy.

Nevertheless, the practice of physical activities in elderly people involves a series of risks and limitations, making it necessary to assume measures that cause said activities to be safer and motivational. This risk can be summarized by the following points [20]:-Avoid exercises that require fast movement that could cause falls.-Avoid exercises that require bad postural execution.-Avoid sudden changes of position and activities that require high-blood levels of stress.-Avoid exercising with closed eyes, due to loss of balance that can be produced at those ages.-Carry out a contingency plan control, heeding external factors related to the activity, like lightning, temperature, etc.-Perform exercises of low intensity.-Do not exceed your own capacity.

To prevent further degradation in the elderly person, it is the key to increase the variety of sensations in the materials used for the physical exercises and improve their sensorial experience [22]. Because of this, it is important to highlight that, in order to reach this wellness improvement, the quality of the physical activities developed is more significant than the quantity of them, assuming an improvement in the participation and execution of physical activities that counteract the loss of vital functions due to ageing.

That way, deeply aware of the internal process that is originated when a stimulus in a person’s organism is picked up, the emotions and intelligences, the psychoevolutives characteristics, the resources employed in this study, and the differentiations between the diverse physical activities, we proceeded with the following study.

## 4. Objectives

Ageing is a complex process, involving genetic, biological, socio-environmental, and cultural factors that show the deterioration of the organism, due to irreversible and common changes found in this stage of human beings. However, this process is individual, as in each person these changes will be different based on biological, physiological, psychological, as well as cognitive aspects. These changes are progressive, but they can also be increasingly delayed to the latest life stage of the human being. That is why living with the best conditions is one of the basic premises intended at this stage of life, including the importance of economic and social factors, as well as the necessity to keep a healthy and active lifestyle. For all this, knowing as much as possible about this age group is vital to attain the most benefits in their practice and with it, improve both quality of life and personal satisfaction.

The objectives posed in this study have been invoked from a generic reflection of the stated research, focused on knowing the existence of the effectiveness of specific sportive material resources to develop exploratory procedures of the sensations in the elderly. Since the materials are not, by themselves, effective, but are self-effective based on the objective, we tried to check their selection and use through the study in order to verify whether they were effective or not in the target achievement that is proposed in each case and in each context type.

From this reflection, it was intended to give complementary responses to other theoretical questions associated with the main subject of our study, such as:-Ascertain possible differences that could have been generated between the characteristics of the specific sportive material resources based on the exploratory procedures of sensations in the elderly.-Find out the emotions awakened by these material resources in the elderly.-Discover how each of the characteristic aspects of the sportive material resources, like size, weight, colour, and texture intervenes in the motivation of elderly people to use these resources when practising physical activities.-To know the most adequate specific sportive material resources for their use in practising sportive physical activities in elderly people to motivate and stimulate satisfying sensations that favour their continued practice of said activities.

## 5. Methodology

### 5.1. Participants

The sample chosen for the study was 365 people between 65 and 100 years old, although the age range that mostly performed physical activities was between 65 and 70 years old, which represents 59.71%. The majority were also women (74.49%), partly because Spanish women have a life expectancy of 85 years compared to 79.2 years for men. This is one of the highest life expectancies in the European Union, a key aspect of the improvement in the quality of life in recent decades [23]. These elderly people all participated in different physical sports activities of physical maintenance, in different sports centres of the Municipal Sports Institute of the City of Seville.

### 5.2. Instrument

Due to the need to analyse different aspects of different scientific fields in this study, the need for both quantitative and qualitative analysis was established, through a test based on a classic research model of quality [24]. To this end, a questionnaire was developed and validated, with eight blocks of closed questions and two open-ended questions to expand and complete the information required.

The questionnaire sought to answer the feelings and emotions evoked in each person on each of the twenty materials presented to them. To avoid misinterpretations, they needed to recognize what was aroused in them on a sheet containing eight photos with elderly people representing the eight emotions, selected by the group of experts from a total of 86 photos previously taken for the study under the same conditions.

### 5.3. Design

The research method used was the Delphy, the group of experts being chosen through a procedure of the aggregation of personal and individual judgments [25]. The procedure was used both for the identification of emotions from the manipulated sports materials and for the evaluation of the positive or negative characteristics of these materials. This method has made it possible to verify and rectify any obstacles before its definitive implementation, ensuring the maximum reliability and validity of the data obtained [26], and leading to a greater agreement between all parties by creating the questionnaire used. This method was also used to validate the photographic template of emotions. This panel consisted of an A3-size panel on which six black and white photographs of elderly people expressing Ekman’s basic emotions [27] (happiness, sadness, fear, anger, disgust, and surprise) were placed, along with a record sheet of the data obtained from the questions asked in the questionnaire.

An assessment was created of the questions asked to the experts to determine whether or not they were fit to continue as experts in the next steps of the questionnaire validation protocol, using an observation tool where each expert rated their level of knowledge on each of the study topics from 1 to 4 (with 1 being the lowest and 4 being the highest) and giving them the opportunity to voice any suggestions. questions, to provide alternative questions to those already elaborated, or to add any others that they considered appropriate [28].

From here, the Knowledge or Information Coefficient (Kc) is easily calculated using the following formula: Kc = n (0.1). Applying this formula, all the experts scored between 0.8 and 0.9, so they were all within the range that qualified them as good. We then conducted a test that allowed each expert to assess aspects that influenced the topic to be studied. These items were: their experience in conducting research; the publication of some of the factors related to the study and their professional experience in some aspects of the study topic. All of the selected experts were considered in all the items at the mean and high values, so all of them were selected as the test experts since none were at the low level. Subsequently, the aspects that influenced the level of the argumentation of the topic to be studied were determined by calculating the argumentation coefficient (Ka) of each expert.

Once the values of the Knowledge Coefficient (Kc) and the Argument Coefficient (Ka) were obtained, we proceeded to obtain the value of the Competence Coefficient (K), which is finally the coefficient that actually determined which experts were considered to work on this research. This coefficient (K) was calculated as follows: K = 0.5 (Kc + Ka), where the Competence Coefficient is (K), the Knowledge Coefficient is (Kc), and the Argument Coefficient is (Ka) [29].

Subsequently, the following selection criteria were established:-High Competition Coefficient: 0.8 < K < 1.0-Average Competence Coefficient: 0.5 < K < 0.8-Low Competition Coefficient: K < 0.5

Finally, ten experts were selected from the initial fifteen, with a profile between 55 and 68 years of age, six women and four men, all of them doctors, of whom four were in Physical Activity and Sport Sciences, two were in Medicine, two were in Education Sciences, and two were in Psychology. Based on these consultations with the selected experts, we enacted specific qualitative changes to the test, always in accordance with what was established by the selected experts. All the contributions of the 10 experts were considered as a brainstorming exercise and, thanks to the clarity and specificity of their comments, there was no need for a second round.

All the items of the questionnaire have been reviewed in detail to adapt them to the target population (elderly people) under the supervision of all the experts, analysing the degree of agreement among them, considering as optimal values those above 0.70, with an agreement interval of 95% among the experts [30].

## 6. Results

More than half of those questioned expressed in our study the fact that they went to practice their physical and sporting activities at the nearest SMI to their home due to their worry about their health (91.1%) followed by those who did it for pleasure (8.12%) (Figure 1).

The results obtained show that the preferred resources are the ones with bigger size, great size (53.78%) or big size (35.76%), which means it is easier to pick up, with a light weight (66.67%), a rough texture (68.09%), which gives them an identification of their sensations, and they curiously preferred darker-coloured resources (68.99%), as they give them a safer and more trusting sensation (Table 1).

Other criteria to highlight the importance and adequate selection of the materials in the sport-oriented physical activities, are those which maximize the sensations that stimulate their sensory organs: sight, hearing, touch, proprioception, etc., distinguishing each of the resources based on the sensation they produced. In such a manner as to the reason that gave them to show each emotion, they have varied, depending on the manipulated material resource in each case, receiving the following results: those which provoked in them anger were not liked (53.93%); equally, those which provoked in them fear were not liked (20.39%); those which provoked disgust were rejected and considered repugnant (67.23%); those which surprised them provoked a sense of uncertainty as they did not know what to expect from them (73.44%); those which provoked happiness were liked and they were attracted to them (83.92%); and those that provoked sadness were rejected as they caused sorrow (11.06%). Therefore, the materials which were liked generally were those that provoked happiness (83.92%), whereas those that did not give any emotion to relate to resulted in indifference (98.92%), and those that were not liked produced anger and rejection (53.93%) (Figure 2).

It was checked that, from the 20 specific material resources chosen, in 18 of them, the most prevalent emotion was happiness, while the other two materials showed surprise. It might be that manipulating these balls reminded them of majorly positive experiences during childhood and youth; whereas the two materials that gave them a surprise emotion (Floorball and Fit ball) declared not knowing them, or not expecting their peculiar characteristics (Table 2).

In order to know the basic motor skills that the questioned elderly preferred to practice with each specific material resource, the results show throwing them, followed by rolling them and hitting them (Table 3).

## 7. Latest Research

As has been mentioned before, this research has no precedent study that links the three central aspects of this paper (elderly people, material resources, and emotions). However, there are recent studies about each of these aspects independently. Although these studies are mainly based on the search for sensation in other segments of the population, they have focused more on the movement between professional life and retirement, as well as the arrival of diseases and the typical fears of the ending of life and not as a search for the stimulus that would cause them to reach positive states [31]. Even the studies performed by Yuste [32] and others about the dilemma between our internal and external environment related to sportive emotions are based on bibliographical reviews on the existing relationship between emotional, physiological, biological, behavioural, and cognitive variables.

Meanwhile, the concept of Emotional Intelligence goes beyond that, since it is a reference both for the education and training of people, as well as their integral development. That is why numerous studies have been undertaken with the intention of linking them with other research variables in order to improve the life quality of the individual in all stages of life, facing daily problems and situations, like their link with other aspects such as:-Stress, commitment, and exhaustion [31].-Anxiety and stress [32,33,34].-Optimism and pessimism [35].-Mental and physical health [36].-Personal satisfaction [37].-Empathy [38].-Quality of life [39].-Support for School Integration [40].

In addition, the term “Emotional Education” is expanding into other areas, finding recent research based on:-School, inclusion, and diversity [41].-Prevention against school bullying [42].-Leadership [43].

Concerning the practice of physical activities, we should mention the research on recreational physical activity and emotional states based on two gerontology groups recorded by Bolaños and Mora [23]. They discovered that, according to the practised physical activity, the emotional statuses can be beneficial, determining that the aquatic medium obtained better results than the terrestrial medium [44].

In recent studies, it is shown that elderly people are not looking to experiment with intense sensations, but to enjoy the known sensations, either through familiar or social links or in the development of activities that causes them wellness. Due to this, the types of activities which characterize this age group are the ones linked to maturity and emotional control, as well as the process of selection, optimization, and compensation, leading to a good performance [45].

Our current study aims to link the result of these investigations together and associate them with the use of material resources specific to their use.

## 8. Discussion and Conclusions

The effectiveness of specific sportive material resources for the development of elderly people must take into consideration that the materials are not effective by themselves but in relation to the objective that is intended to be achieved with their use. Thus, the conclusion of this study is that this effectiveness depends on the feelings that their use elicits in these elderly people. The choice of materials to be used leads to greater motivation and continuity in the practice of physical activities and thus improves their quality of life. Based on this premise, the most important findings of the study were the following:-There are differences in the procedures when exploring the sensations of elderly people who engage in physical activities. They vary depending on the characteristics of these resources, with materials of large size to facilitate their reception, light weight, rough texture to facilitate their identification, and dark colour to give them a greater sense of security being the most accepted.-The exploratory procedure of sensations of specific sports material resources is influenced by previous experiences lived by each person.-Each specific material resource awakens different emotions depending on the different characteristics of each material: weight, size, texture, and colour.-The most appreciated spherical material resources for practising physical activities in general are those that provoke the emotion of happiness.-The motor skills that elderly people like to perform with spherical materials are throwing, followed by rolling, and hitting.

These findings must be taken into account when selecting the materials that the elderly community will use in their physical activities so that technicians can plan fun and motivating sessions. The aim should be to provide them with continuity in the practice of an active life, setting healthy habits to achieve a better quality of life for much healthier ageing.

## Figures and Tables

**Figure 1 ijerph-19-11775-f001:**
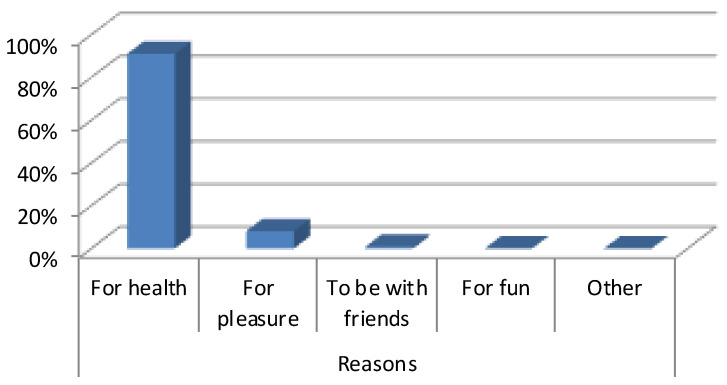
Motivation for physical activity of elderly (Own production).

**Figure 2 ijerph-19-11775-f002:**
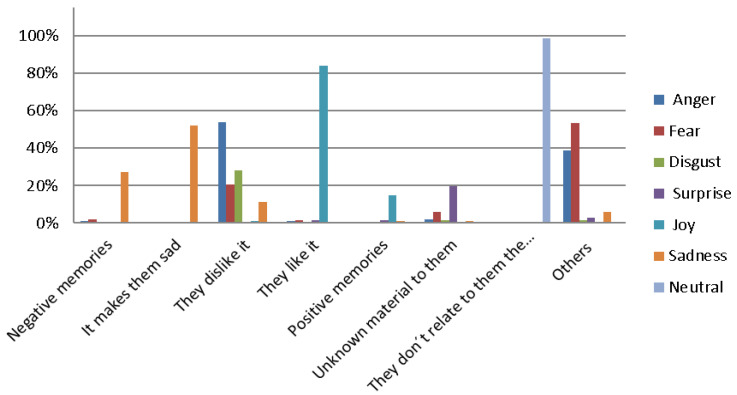
Emotions produced by materials in the elderly (Own production).

**Table 1 ijerph-19-11775-t001:** Materials preferred by the elderly according to their characteristics (Own production).

Size	%	Weight	%	Texture	%
Very big	53.78	Lightweight	66.67	Smooth	34
Big	35.76	Heavy	33.33	Rough	66
Medium	9.01				
Small	1.45				

**Table 2 ijerph-19-11775-t002:** Emotions according to each material (own production).

Materials	Anger	Fear	Disgust	Surprise	Joy	Sadness	Neutral
Table Tennis Ball	17.97%	2.61%	3.77%	5.80%	60.58%	3.19%	6.09%
Bowling	4.93%	8.41%	5.80%	18.84%	55.36%	2.61%	4.06%
Golf ball	6.67%	6.96%	5.80%	8.70%	48.41%	16.23%	7.25%
Pool ball	7.27%	4.94%	9.30%	9.30%	39.53%	9.59%	20.06%
Fronton Ball	8.12%	2.61%	5.51%	7.54%	49.86%	22.90%	3.48%
Tennis ball	4.07%	2.33%	10.47%	2.33%	76.74%	1.74%	2.33%
Petanque Ball	6.09%	6.09%	6.38%	4.35%	54.78%	17.10%	5.22%
Rhythmic Gymnastics Ball	2.32%	0.87%	1.74%	6.09%	66.96%	3.19%	18.84%
Volleyball ball	2.90%	5.22%	2.32%	5.22%	70.72%	5.22%	8.41%
Basketball ball	6.09%	4.64%	2.61%	5.80%	70.43%	4.35%	6.09%
Cork ball American	2.32%	0.58%	4.35%	28.99%	39.42%	4.06%	20.29%
Hacky ball	3.77%	0.00%	3.48%	21.74%	62.61%	3.77%	4.64%
Cork ball	1.16%	1.74%	17.10%	26.96%	34.49%	5.80%	12.75%
Massage ball	11.88%	2.03%	5.51%	8.70%	55.65%	10.72%	5.51%
Floorball ball	3.49%	1.45%	2.03%	39.24%	37.21%	3.49%	13.08%
Juggling ball	4.06%	2.61%	4.06%	7.54%	74.49%	4.06%	3.19%
Small tennis ball	4.35%	5.51%	20.00%	5.22%	57.10%	3.19%	4.35%
Rubber ball	2.03%	5.51%	13.91%	4.64%	68.70%	1.74%	3.48%
Small basketball ball	0.87%	0.87%	0.87%	6.38%	87.54%	2.90%	0.58%
Fitball	0.58%	5.22%	2.90%	41.74%	39.13%	9.28%	1.16%

**Table 3 ijerph-19-11775-t003:** Motor actions that the elderly like to perform with each material (own production).

Materials	Throw	Bounce	Hit	Roll	Smash	Other	None
Table Tennis Ball	35.36%	33.04%	27.83%	2.03%	0.29%	0.29%	1.16%
Bowling	61.74%	0.00%	10.72%	24.64%	0.87%	0.58%	1.45%
Golf ball	23.77%	0.87%	60.87%	10.14%	0.29%	2.61%	1.45%
Pool ball	25.51%	0.00%	19.71%	37.97%	1.16%	14.78%	0.87%
Fronton Ball	26.67%	1.45%	63.77%	4.35%	2.32%	1.16%	0.29%
Tennis ball	37.39%	12.46%	45.22%	2.03%	0.58%	2.32%	0.00%
Petanque Ball	57.97%	1.45%	5.51%	26.38%	0.87%	7.25%	0.58%
Rhythmic Gymnastics Ball	52.75%	25.51%	6.67%	8.70%	3.77%	1.74%	0.87%
Volleyball ball	33.33%	23.48%	36.81%	4.35%	0.29%	1.45%	0.29%
Basketball ball	16.81%	64.06%	2.03%	2.61%	0.29%	13.04%	1.16%
Cork ball American	40.29%	1.45%	4.93%	14.49%	15.36%	11.59%	11.88%
Hacky ball	17.68%	0.29%	5.51%	2.90%	64.93%	4.06%	4.64%
Cork ball	33.91%	0.29%	6.38%	25.22%	13.04%	10.14%	11.01%
Massage ball	14.78%	0.29%	3.48%	30.43%	33.33%	14.20%	3.48%
Floorball ball	52.17%	0.58%	21.45%	7.25%	2.61%	10.14%	5.80%
Juggling ball	59.13%	0.29%	6.38%	2.61%	25.80%	4.93%	0.87%
Small tennis ball	33.62%	31.01%	23.48%	6.38%	2.90%	1.74%	0.00%
Rubber ball	26.09%	20.58%	8.70%	12.75%	31.01%	0.87%	0.00%
Small basketball ball	16.81%	75.65%	1.74%	3.19%	0.58%	1.74%	0.29%
Fitball	10.43%	5.22%	1.74%	30.14%	29.86%	19.13%	3.48%

## Data Availability

Not applicable.

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
