# Peer review of "Emotional Response in the Elderly to Sports Materials"

_ijerph, 2022, doi:10.3390/ijerph191811775_

Round 1
Reviewer 1 Report
The topic is original and does not seem to have been studied before.
It is not clearly explained in the work whether, depending on the objectives of the physical activity in question, one type of material may be better than another. It speaks of "physical activities" in general, but depending on the objective of these activities, one type of material or another will be more appropriate.
It is necessary to include a reference to support the data in lines 25-27. In addition, it would be interesting to include a broader view of this topic, not just Spain.
It needs a revision of the English language as there are sentences that are not well understood (for example, lines 371-372)
There is mention a "group of experts" (line 384) but it is not specified how many there are and their characteristics.
The figures should each have a title (see figure 2) and include the percentages so that they can be interpreted directly without reading the text.
Figures 4-5 cannot be interpreted correctly. Wouldn't it be better to include them in table format?
There is no real discussion and the conclusions are rather a summary of the results put in a schematic form.
The references used are dated before 2015 (when the results of the study were collected). This seems to indicate that a literature review has not been carried out after this date and that it should be carried out in order to know the possible advances produced in the last 7 years.
Author Response
Dear Reviewers:
Thank you very much for your invaluable work in trying to improve our paper. All your indications have been corrected, as well as those of the other reviewer, so we have rewritten the text as you suggested. The changes are the following:
- We have made a new revision of the language. We have modified the grammatical aspects that you indicated by hiring a company of native translators.
- We have incorporated more details into the methodology and replaced some heavily loaded figures with tables that, allowing allow greater clarity when reading and greater scientific rigour.
- We have included some more recent quotes but not many due to the scarce investigation related to our research, which links the four fields of study (older people, physical activity, material resources and emotions).
- As for the suggestion of one of the reviewers to include a broader vision of this topic to other countries outside of Spain, we state that after the bibliographic review carried out, not only in Spanish, but also in French and English, we see that the investigations have been based on each of the study factors separately, but in none of them are the four areas studied at the same time. Nevertheless, our current Research Group is trying to connect a network of groups from different European and Ibero-American countries, in order to carry out this suggestion. With this, we hope that its development can provide new and different broader studies on the subject in the future.
Therefore, we refer to you the new manuscript submitted with all the modifications requested. We hope you like it, and we are at your disposal for any changes you may consider.
Please, see the attached file.
Thanks beforehand for your time.

Reviewer 2 Report
Overall: This paper has good potential since the subject matter is important. As it is, it should be restructured and rewritten to provide greater context, references, and organized in a more traditional order to help the reader follow the thought patterns and methodology.
Abstract: The abstract needs to be broken down into shorter sentences and provide more background information to contextualize the study.
Line 14-15 – This is an awkward sentence, please consider rephrasing since it is difficult to understand in its current state.
Introduction:
The concepts and background information seems good. There are significant grammatical errors (only some of which are listed below) and a lack of citations for the information provided.
Line 21 – should be “have” instead of “has”
Line 23 – it is unclear as to what has been increasing
Line 24 – this sentence is awkward, consider rephrasing.
Line 25 – Do you mean 5.3% OF the total population?
Line 29 – it is unclear what specifically will “quicken”.
Almost every sentence in the introduction should be checked for grammar.
Rationale:
This section also needs significant edits to correct grammar.
Line 86 – there are a lot of elements in this paragraph that are summarized in the last sentence. The reference here should be more specific and additional references should appear earlier in the section where other information is presented.
Line 98-100 – What, specifically is being referenced here? The references appear with the name but no information.
Physical Activities and Aging:
The rationale for the materials chosen should be part of the introduction and rationale sections to definitively tie them to the emotional development and intelligence discussed. The connection is unclear.
The Latest Researchers section should be included before the discussion about the research setup to help tie these together.
The arrangement of this paper should be changed so that it is clear that it is a scientific study with a solid introduction/literature review that provides context and existing studies for each part of this study all in one section followed by materials and methods and then the results. As it is right now, the paper is difficult to follow and the methodology is hard to understand. I recommend a full rewrite to clarify the purpose and context of the study, detailed separate methodology and data sections (with clear participant sections) followed by results that can be easy to understand based on the previous sections. I applaud the desire for mixed method studies, but it needs to clarified in a methods section.
Author Response
Dear Reviewer:
Thank you very much for your invaluable work in trying to improve our paper. All your indications have been corrected, as well as those of the other reviewer, so we have rewritten the text as you suggested. The changes are the following:
- We have made a new revision of the language. We have modified the grammatical aspects that you indicated by hiring a company of native translators.
- We have incorporated more details into the methodology and replaced some heavily loaded figures with tables that, allowing allow greater clarity when reading and greater scientific rigour.
- We have included some more recent quotes but not many due to the scarce investigation related to our research, which links the four fields of study (older people, physical activity, material resources and emotions).
- As for the suggestion of one of the reviewers to include a broader vision of this topic to other countries outside of Spain, we state that after the bibliographic review carried out, not only in Spanish, but also in French and English, we see that the investigations have been based on each of the study factors separately, but in none of them are the four areas studied at the same time. Nevertheless, our current Research Group is trying to connect a network of groups from different European and Ibero-American countries, in order to carry out this suggestion. With this, we hope that its development can provide new and different broader studies on the subject in the future.
Therefore, we refer to you the new manuscript submitted with all the modifications requested. We hope you like it, and we are at your disposal for any changes you may consider.
Please, see the attached file.
Thanks beforehand for your time.

Round 2
Reviewer 1 Report
Authors have made a significant improvement from the previous version of the text, especially the methodology and the results section. Concerning the language, the improvement has also been significant.
I see only minor changes in the results section, in particular:
- Figure 2: not include 120% and avoid decimals on the y-axis
- Tables 2 and 3: group the type of material in alphabetical order, grouping together those of similar origin (e.g. basketball ball (standard) / basketball ball (small)).
Author Response
I thank you for your evaluation of our changes and I would like to point out that with respect to figure 2, the percentages have been modified as you indicate.
The grouping of the materials has not been possible. Because when working with two basketballs, one is the official Basketball ball (24 cm in diameter), and the other ball is a small rubber basketball (3 cm in diameter), which imitates the Basketball; For this reason, we have established two groups of materials: the approved ones and the alternative ones. And each of those balls correspond to one of those different groups. Hoping our answers are satisfactory for you.
Thanks a lot.

Reviewer 2 Report
Dear Authors,
Thank you for taking the time to revise your manuscript. The edits have made a huge difference in the ability to convey your methods and findings.
Author Response
Thank you for your consideration of the changes made and your contributions to improve the manuscript.
Thanks a lot.
